# A deep learning approach to segmentation of the developing cortex in fetal brain MRI with minimal manual labeling

Ahmed E. Fetit[1]                                     AFETIT@IMPERIAL.AC.UK
Amir Alansary[1]                                A.ALANSARY14@IMPERIAL.AC.UK
Lucilio Cordero-Grande[1]                  LUCILIO.CORDERO_GRANDE@KCL.AC.UK
John Cupitt[1]                                    J.CUPITT@IMPERIAL.AC.UK
Alice B. Davidson[2]                          ALICE.B.DAVIDSON@KCL.AC.UK
A. David Edwards[2]                             AD.EDWARDS@KCL.AC.UK
Joseph V. Hajnal[2]                              JO.HAJNAL@KCL.AC.UK
Emer Hughes[2]                                 EMER.HUGHES@KCL.AC.UK
Konstantinos Kamnitsas[1]        KONSTANTINOS.KAMNITSAS12@IMPERIAL.AC.UK
Vanessa Kyriakopoulou[2]              VANESSA.KYRIAKOPOULOU@KCL.AC.UK
Antonios Makropoulos[1]                          BENUIX@GMAIL.COM
Prachi A. Patkee[2]                           PRACHI.PATKEE@KCL.AC.UK
Anthony N. Price[2]                          ANTHONY.PRICE@KCL.AC.UK
Mary A. Rutherford[2]                    MARY.RUTHERFORD@KCL.AC.UK
Daniel Rueckert[1]                          D.RUECKERT@IMPERIAL.AC.UK

[1] *Biomedical Image Analysis Group, Department of Computing, Imperial College London, London, SW7 2AZ, United Kingdom.*
[2] *Centre for the Developing Brain, Division of Imaging Sciences and Biomedical Engineering, King's College London, London, SE1 7EH, United Kingdom.*

## Abstract

We developed an automated system based on deep neural networks for fast and sensitive 3D image segmentation of cortical gray matter from fetal brain MRI. The lack of extensive/publicly available annotations presented a key challenge, as large amounts of labeled data are typically required for training sensitive models with deep learning. To address this, we: (i) generated preliminary tissue labels using the *Draw-EM* algorithm, which uses Expectation-Maximization and was originally designed for tissue segmentation in the neonatal domain; and (ii) employed a human-in-the-loop approach, whereby an expert fetal imaging annotator assessed and refined the performance of the model. By using a hybrid approach that combined automatically generated labels with manual refinements by an expert, we amplified the utility of ground truth annotations while immensely reducing their cost (283 slices). The deep learning system was developed, refined, and validated on 249 3D T2-weighted scans obtained from the *Developing Human Connectome Project*'s fetal cohort, acquired at 3T. Analysis of the system showed that it is invariant to gestational age at scan, as it generalized well to a wide age range (21 – 38 weeks) despite variations in cortical morphology and intensity across the fetal distribution. It was also found to be invariant to intensities in regions surrounding the brain (amniotic fluid), which often present a major obstacle to the processing of neuroimaging data in the fetal domain.

**Keywords:** Fetal, developing, brain, cortex, gray matter, 3D segmentation, deep learning.

## 1. Introduction

During early stages of human brain development, the cortex undergoes rapid transformations in texture and morphology; expanding from a smooth, homogeneous composition into a dense, highly convoluted structure (Figure 1). Throughout this process, cellular connections start to form between distant brain regions, mapping out the foundations for humans' advanced cognitive abilities. Alongside this, functional activations and connections also start to develop across the brain; a crucial process for the development of human cognition (Van Essen, 1997; Doria et al., 2010; Ball et al., 2013; Makropoulos et al., 2018b). Work by Turk and colleagues (Turk et al., 2019), and efforts such as the *Developing Human Connectome Project* [1] *(dHCP)* (Makropoulos et al., 2018b; Bastiani et al., 2019; Fitzgibbon et al., 2019) and the *Baby Connectome Project* [2] (Howell et al., 2019) aim to advance magnetic resonance imaging (MRI) techniques in order to map out such structural and functional changes in early brain development. Such mapping, also known as the connectome, would allow the neuroscience community to investigate neurobiological mechanisms, as well as genetic and environmental factors that underpin healthy brain development in fetuses and neonates. It could also help understand the development of neurological conditions such as cerebral palsy and autism.

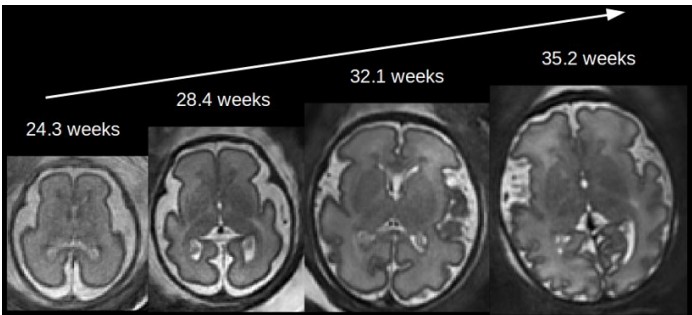

Figure 1: Example T2-weighted axial slices of four different fetal subjects from the *dHCP* cohort, imaged at various gestational ages: 24.3, 28.4, 32.1, and 35.2 weeks. One can observe striking changes in cortical gray matter intensity and morphology taking place within narrow time frames. This makes image segmentation of the developing cortex from the fetal brain MRI a tremendously challenging task.

An important step towards delivering an accurate and representative connectome is robust structural processing (e.g. tissue segmentation) of brain MRI scans; a particularly challenging task when carried out on perinatal data sets[3]. As discussed by Makropoulos and colleagues (Makropoulos et al., 2018b), visual characteristics of fetal and neonatal brains are significantly different from those of adult brains in terms of size, morphology, and white/gray matter intensities. Such changes in intensities during brain development are a result of the

---

1. developingconnectome.org

2. babyconnectome.org

3. *Perinatal* here refers to relatively short periods before and after birth *à la* both fetal and neonatal data sets. Perinatal scans acquired by the *dHCP* consortium range from 20 to 45 weeks of gestational age. In this work, only scans from the *dHCP* fetal cohort were included (gestational age at scan: 21 - 38 weeks).

continuous decrease of water content within the brain, as well as the formation of myelin sheaths around white matter tracts, i.e. myelination (Serag, 2013). Additionally, scanning times tend to be limited to ensure the comfort of mothers and babies, causing the spatial and temporal resolutions of the images to be much lower and more prone to motion artifacts compared to those acquired from adults. Hence, whilst a plethora of methods is available for use on adult data sets, the analysis of developing brain MRI needs to be addressed as a separate problem. The translation of image segmentation methods optimized for use on adult data into the perinatal domains remains a tremendously challenging task.

## 2. Related Work

### 2.1. Conventional segmentation techniques

In light of the above, Makropoulos and colleagues presented an automated pipeline for processing structural scans obtained from neonates (Makropoulos et al., 2018b). The pipeline's workflow comprised a number of steps that included bias correction, brain extraction, tissue segmentation, brain parcellation, white-matter mesh extraction, and cortical surface reconstruction. The tissue segmentation aspect of the workflow incorporated the well-established *Draw-EM* (Makropoulos et al., 2014); an open-source software for neonatal brain MRI segmentation based on the Expectation-Maximization algorithm. Readers are referred to a review by (Makropoulos et al., 2018a) for a thorough survey of the image segmentation literature in perinatal MRI. Notably, a plethora of the structural processing methods developed in the neonatal domain focused on image segmentation brain tissues. Examples systems include MANTiS (Beare et al., 2016) and Neoseg (Prastawa et al., 2005), as well as various works reported by (Anbeek et al., 2008, 2013; Srhoj-Egekher et al., 2012; Chiţă et al., 2013; Moeskops et al., 2015; Wang et al., 2015; Sanroma et al., 2016). Within the context of fetal imaging, studies looked into the application of parametric methods (Habas et al., 2008, 2009, 2010; Ison et al., 2012; Wright et al., 2014) as well as deformable modeling techniques (Dittrich et al., 2011; Gholipour et al., 2012).

### 2.2. Techniques based on machine learning

In terms of machine learning methods, a number of supervised classification algorithms were explored in the literature, these include random forests, e.g. (Ison et al., 2012; Kainz et al., 2014), support vector machines (Keraudren et al., 2014), and more recently, deep neural networks (Moeskops et al., 2016; Rajchl et al., 2016; Khalili et al., 2019). Moeskops and colleagues were amongst the first to study the application of deep learning for brain MRI tissue segmentation; an important step in neuroimaging structural processing workflows, and one that is also needed to localize functional and diffusion signals from labeled anatomical structures. Their work showed accurate segmentation as well as spatial consistency for developing brains (namely preterm neonates), as well as young and ageing adult brains. The recent work by Khalili and colleagues focused on the application of deep learning on the specific problem of brain tissue segmentation from fetal MRI (Khalili et al., 2019). Brain tissue segmentation can be particularly difficult to carry out on fetal scans due to the aforementioned rapid changes in shape and intensity that take place over narrow time

scales. As a result, it is very challenging to optimize an individual algorithm to be generic enough for use on scans obtained from various fetal age groups.

## 3. Contribution and overview

In this regard, and as a step towards producing a robust map of the fetal connectome, we developed a novel, age-invariant system for fast and sensitive 3D image segmentation of cortical gray matter from the developing fetal brain. The system is based on the deep learning family of algorithms (LeCun et al., 2015), namely convolutional neural networks (CNNs), and was designed to work on volumetric, T2-weighted brain MRI scans. A major constraint to using deep learning techniques is the need to collect large, manually labeled ground truth annotations; a tedious task that can take months to complete by an experienced annotator. Image annotation in the fetal domain poses the additional challenge that data is highly heterogeneous; brain morphology and intensity characteristics change rapidly throughout development. To address this bottleneck, we leveraged the *Draw-EM* algorithm, which was originally developed for the neonatal domain, and used it to generate preliminary 3D labels in place of ground truth annotations. We then employed a human-in-the-loop approach, whereby an expert fetal imaging annotator assessed and refined the performance of the model using fewer than 300 2D slices, substantially cutting down the cost of ground truth annotations. A key novelty in our approach is in updating the complex 3D segmentations with corrections made only on a small number of selected 2D slices, sampled from axial, coronal, and sagittal directions. Our system was developed, refined, and evaluated using a large number of 3D scans (n=249, gestational ages= 21-38 weeks), acquired at 3T. Our analysis of the system showed that it is invariant to the changes in shape and intensity which occur during early brain development; such heterogeneity often impedes the development of segmentation systems that can generalize across the fetal ages.

## 4. Materials and Methods

Figure 2 visually depicts our methodology. Details are discussed below.

### 4.1. Image acquisition

A total of 249 T2-weighted 3D scans were obtained retrospectively from the *dHCP* fetal cohort. Acquisition was carried out by the *dHCP* consortium using a Turbo Spin Echo (TSE) sequence on a 3T Philips scanner, following a protocol described by (Price et al., 2018). Details of image acquisition are available in Appendix A.

### 4.2. Addressing the ground truth bottleneck

*i) Generating preliminary 3D training labels using Draw-EM*
The current gold standard for image segmentation is manual delineation (Gousias et al., 2012); achieving robust segmentation via a deep learning set-up would conventionally require the presence of large amounts of ground truth labels carried out by experienced annotators. We addressed this by capitalizing on labels generated by *Draw-EM* (Makropoulos et al., 2014), originally designed for neonatal MRI. Our goal was to automatically obtain effective

groupings of tissues into semantic classes, one of which is cortical gray matter, and to use them as preliminary labels for training an initial deep segmentation model. Labels generated included: background, brainstem, cerebrospinal fluid, deep gray matter, germinal matrix, cortical gray matter, outliers, ventricles, and white matter (Appendix B).

The number of 3D scans available was 249; 151 of those were processed by *Draw-EM*. Since this variant of *Draw-EM* was not extensively validated on **fetal** scans prior to use on our cohort, it was plausible that it might not have performed sufficiently well on all scans. The output of *Draw-EM* therefore visually inspected by a postdoctoral researcher who grouped the scans into two categories - pass and fail. Scans in the pass category were those successfully registered to the developing brain atlas and had segmentation labels assigned into reasonable regions of the brain, regardless of sensitivity. The remaining scans were assigned the fail category; those were predominantly scans that failed the registration step of *Draw-EM*. The 98 scans not processed by *Draw-EM* were retained for use in subsequent steps: 12 scans were retained for the clinical refinement step (*held-out-set-B*), while 86 scans were retained for ultimate assessment after development and refinement (*held-out-set-A*).

### ii) Initial data separation

Of the 151 scans processed by *Draw-EM*, 92 were assigned the pass category in the manual quality control step, and were further subgrouped into two sets. The first subgroup was the *model-development-set*; 49 T2-weighted scans were included with the purpose of developing and optimizing a deep neural network capable of segmenting developing brain tissues, based on their visual characteristics. Of those 49 scans, 39 were used for model training and 10 were used for validation throughout the training cycle. Subjects' GAs at scan in this set ranged between 22.4 and 38 weeks. The second sub-group comprised 43 scans; these were completely retained from the model during training (hereby referred to as the *held-out-set-C*). This sub-group was retained for the purpose of providing an initial assessment of the model's performance. GAs in the held-out set ranged between 23.3 and 36.5 weeks.

### iii) Training a preliminary 3D tissue segmentation network

Scans were normalized to ensure zero mean and unit variance intensities (bias field correction was not carried out). A deep 3D network was trained on the *model-development-set* using *DeepMedic* v0.7.0 (Kamnitsas et al., 2017) to classify voxels into one of the 9 tissue classes. Labels generated by *Draw-EM* were used in place of ground-truth annotations. Throughout the training cycle, 39 scans were used for training and 10 were used for validation. This version of the model is referred to as *Segmentation Network 1 (SN1)*.

The adopted 3D architecture is based on multiple parallel convolutional pathways, each processing the input at different scales (resolutions). Features encoded by the pathways are complementary, creating a feature hierarchy suitable for the task of tissue segmentation: pathways processing higher resolution input encode local, detailed features, which are required for precise segmentation; pathways processing low-resolution input capture higher-level contextual information, such as for improving localization of structures (Kamnitsas et al., 2017). In this work, we used three parallel pathways. The first processes input at normal resolution, while the other two operate at resolutions down-sampled by 3 and 5 times, respectively. All pathways comprised 8 layers of [3,3,3] kernels. Output features are then concatenated and processed by a final classification layer. The training cycle for

this preliminary network comprised 35 epochs, each consisting of 20 sub-epochs. In every sub-epoch, images were loaded from 50 cases, and 1000 segments were extracted in total. Training batch size was set to 5. Initial learning rate was set to 0.001 and was halved at pre-defined points using a scheduler (epochs 17, 22, 27, 30 and 33). Changing the mean and standard deviation of training samples was carried out in order to augment the dataset. Training was accelerated using an NVIDIA Tesla K80 graphics processing unit (GPU).

### iii) Clinical refinement of preliminary cortical segmentation samples

An experienced fetal brain imaging annotator was presented with segmentation maps produced by the initial model *SN1*. Slices were obtained from a selection of 101 volumes representative of the age distribution within the cohort. These volumes were sampled from the 151 scans processed by *Draw-EM*, as well as the 12 scans in *held-out-set-B*, ensuring variability is introduced into the selection. For each volume, the annotator was presented with the mid axial, sagittal and coronal slices together with their corresponding segmentation labels predicted by *SN1*. The use of slices from the three different plane views was done to ensure various aspects of the brain anatomy were sufficiently represented within each volume. Slices were sampled from the centers of the volumetric scans; variations in brain sizes as well as in regions surrounding the brain within each scan ensured that the extracted slices were not from the exact centers of the brains. This was done to increase the range of visual characteristics presented to the annotator. We aimed to extract 3 slices from each of the 101 volumes. This initially resulted in 303 slices, and a visual check reduced the number of slices of sufficient quality to 283 (101 axial, 86 coronal, and 86 sagittal). Using a clinically-established annotation protocol, segmentations were refined by the annotator on the slices. By using a hybrid approach that combined automatically generated labels with manual refinements, we aimed to amplify the utility of ground truth annotations and immensely reduce their cost to those 283 2D slices.

### 4.3. Formulating a 3D cortical segmentation network and final evaluation

The segmentation task was reformulated as a binary problem: **cortex vs. non-cortex**, and a new deep network was trained on the *model-development-set*, also using *DeepMedic* v0.7.0. The cortical segmentation network was trained as per the settings discussed above, also on the *model-development-set*, albeit with a number of variations. Training was carried out over a longer period of 65 epochs. Learning rate halving schedule was set to more frequent pre-defined points: epochs 17, 22, 27, 32, 37, 42, 47, 52. Finally. a more extensive data augmentation strategy was used, whereby 11 different combinations of Gaussian distribution mean and standard deviation values were used to introduce sampling variations. The architecture of the network was, however, left unchanged. The resulting model is referred to as *Segmentation Network 2 (SN2)*. Refined cortical gray matter labels produced on the *dHCP* 3T slices by the expert annotator were incorporated into *SN2* (243 training, 30 validation), which was fine-tuned over a new training cycle that comprised 50 epochs. The state of the trainer was, however, reset at the beginning of each session. Learning rate scheduling and data augmentation strategies were carried out as discussed above. The final system was applied on *held-out-set-A* which consisted of 86 full *dHCP* volumes.

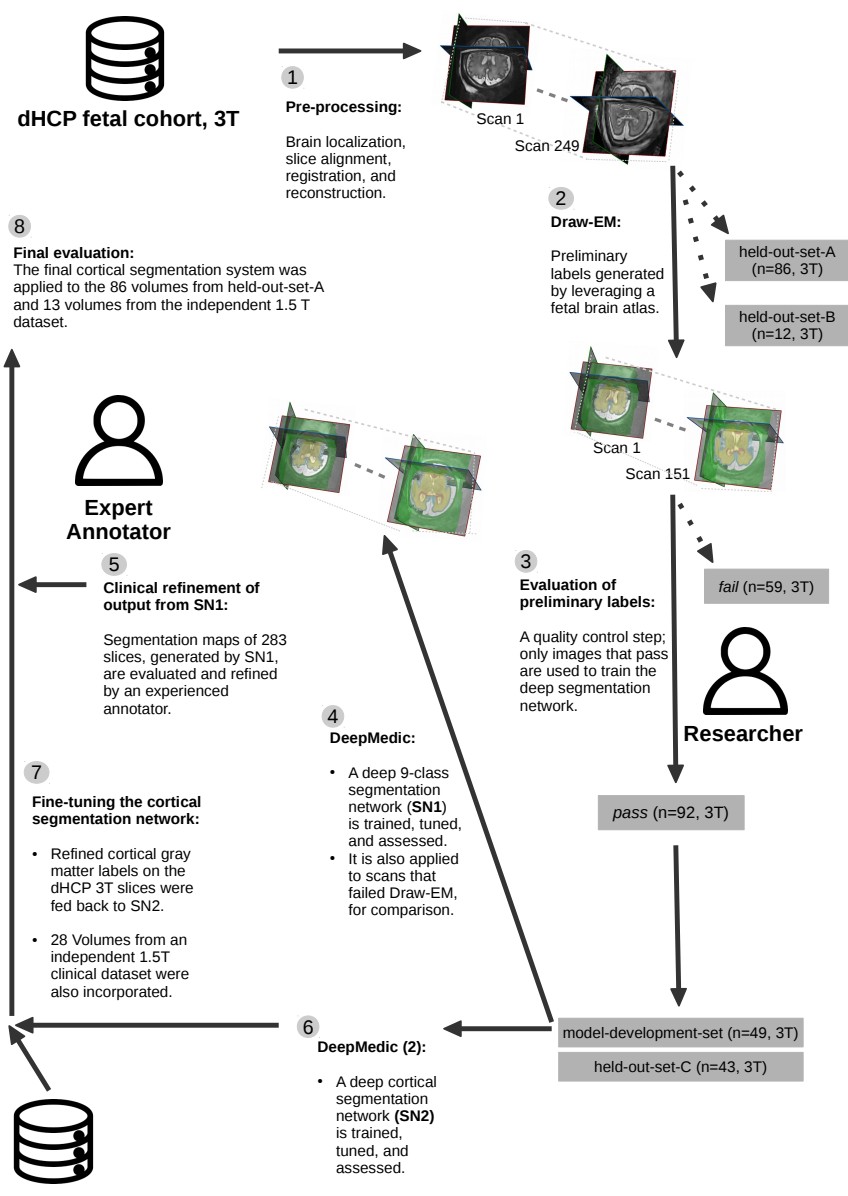

Figure 2: Workflow taken to develop, refine, and validate the cortical segmentation system with minimal manual labeling through a human-in-the-loop approach. We have released the final CNN model, in addition to videos showing example segmentation performance on completely unseen 3D scans, publicly on Github.

### 4.4. Studying transferability into a 1.5T distribution

To explore transferability of *SN2* on fetal scans from a different domain distribution, the above process was repeated, albeit with an additional 41 full volumes from an independent 1.5T clinical cohort. Details of a preliminary experiment can be found in Appendix D.

## 5. Results and Discussion

### 5.1. Preliminary training labels generated with *Draw-EM*

92 of the 151 scans were successfully registered and had acceptable tissue labels assigned to corresponding brain regions. Visual inspection of the remaining 59 scans indicated that failures were predominantly due to errors in registration; these resulted in, for instance, regions outside the brain wrongly assigned brain tissue labels of similar intensity ranges.

### 5.2. Development and assessment of initial model *SN1*

The initial segmentation model *SN1* was successfully trained to capture discriminative patterns for the segmentation of tissues in fetal brain MRI. The model at convergence was applied to *held-out-set-C* (n=43) which was completely retained during training. As per Appendix C, quantitative assessment showed good performance, with cortical gray matter segmentation showing a DSC of 76% (standard deviation = 8.5%). DSC values obtained on the remaining tissue classes ranged between 80% and 90% (with the exception of the germinal matrix class which achieved a relatively low score of 56%). The observed sensitivity, specificity, and positive predictive values were similarly very high. It remains noteworthy that these reported metrics were computed by leveraging the output of *Draw-EM* in place of ground-truth labels. Visual inspection of the output maps was carried out to provide additional reassurance; the obtained segmentation delineated tissue groupings into 9 classes that represented the brain anatomy sufficiently well. An interesting observation was that upon training *SN1* using labels generated from *Draw-EM*, *SN1* was actually able to segment data in both the pass (n=92) and fail categories (n=59). Discussion of results from subsequent steps focuses on cortical gray matter; the primary interest of this paper. The processes described on cortical gray matter are parts of a substantially bigger project, and are being replicated with the remaining tissue classes using labels generated by *SN1*.

### 5.3. Clinical refinement of cortical segmentation output from *SN1*

Figure 3 illustrates a sagittal slice (GA at scan 27.5 weeks) from the sample presented to the expert annotator, in addition to corresponding cortical gray matter segmentations generated by *SN1*, as well as subsequent refinements that were manually carried out. The figure shows example over-segmentation errors that occured by the *Draw-EM* trained network, where voxels forming boundary regions between cortex and surrounding tissues required corrections and improvements.

### 5.4. Developing, validating and evaluating *SN2* for 3D cortical segmentation

At the end of the 50-epoch training cycle, the cortical segmentation model at convergence achieved mean and median DSC values of 0.82 and 0.84 respectively on the validation set

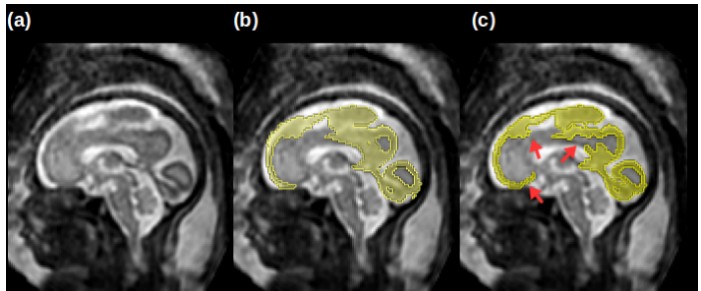

Figure 3: (a) Sagittal slice from scan acquired at GA of 27.5 weeks. (b) Corresponding cortical gray matter segmentation generated by initial model *SN1*. (c) Manual refinements carried out by the annotator to address over-segmentation.

(standard deviation = 0.07), which consisted of 2D slices refined by the expert annotator (n=30). The slices corresponded to subjects scanned between a wide range of GAs (22.2 to 37.3 weeks) and comprised axial, sagittal and coronal views. We applied the model to *held-out-set-A* (full volumes, n=86) to evaluate its performance on completely retained 3D data. All evaluation was done in 3D. Figures 4-6 show representative images (axial view) illustrating various stages of cortical development, in addition to segmentation masks generated by the system; the segmentation shown is representative of the results observed across the held-out set. The results indicate that the system is age-invariant, in the sense that accurate and sensitive segmentation maps were successfully produced for scans sampled from a wide range of gestational ages (e.g. 27, 31, 32.4 week scans in Figures 4-6).

The results also indicate that the system is invariant to changes in intensity across structures surrounding the brain (amniotic fluid), and does not require separate brain detection/ region-of-interest masks prior to cortical segmentation; a time consuming pre-processing step that is common in neuroimaging studies. Noteworthy, the system is computationally efficient, with complete segmentation of the 86 volumes taking only 1,441 seconds (24 minutes) on an NVIDIA Tesla K80 GPU device. Finally, whilst a pre-processing step was carried out to roughly align the scans to standard pose, this does not need to be carried out perfectly as the segmentation remains efficient regardless of perturbations in initial brain orientation, as per Figures 4-6. For detailed visualization of 3D segmentation examples, readers are referred to our publicly available Github repository and example videos. The repository also includes a separate brain region detection network that can be used to provide ROIs for use alongside the cortical segmentation CNN.

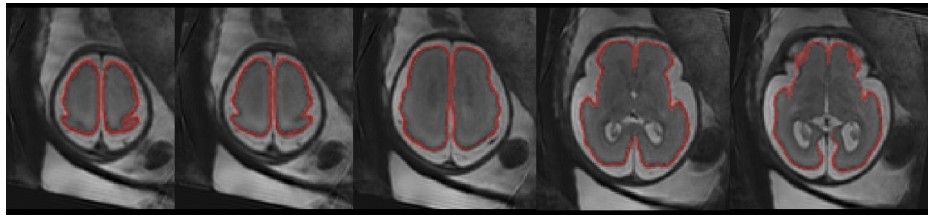

Figure 4: Performance of *SN2* on a completely held-out 3D scan, GA: 27 weeks.

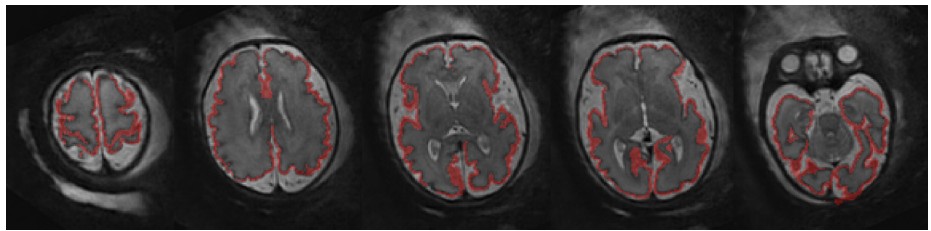

Figure 5: Performance of *SN2* on a completely held-out 3D scan, GA: 31 weeks.

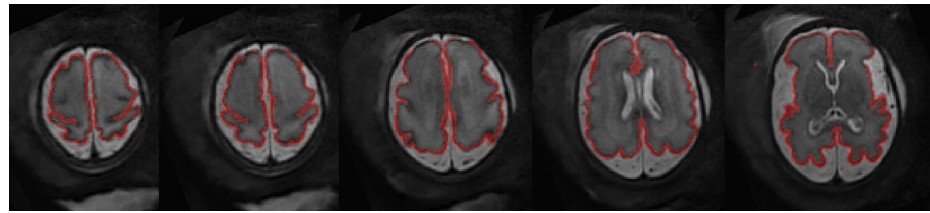

Figure 6: Performance of *SN2* on a completely held-out 3D scan, GA: 32.4 weeks.

## 6. Conclusion and future work

We used a human-in-the-loop-approach to develop a fast and sensitive 3D cortical segmentation system based on deep neural networks, without the need for large manual annotations. Our approach can be adopted in a wide range of neuroimaging applications, where the need for vast manual input from domain experts can impede rapid algorithm development and implementation. It remains noteworthy, however, that the availability of *Draw-EM*, originally designed for neonatal MRI, provided a good starting point for the specific problem of fetal MRI segmentation; a potential constraint in other applications. We therefore ought to explore other machine learning paradigms in the future, e.g. active learning, where expert annotators only label scans which are estimated to be of most potential value to the model. Finally, the segmentation work described in this paper on cortical gray matter is part of a bigger project and is being replicated for the other tissues using labels generated by *SN1*.

### Acknowledgments

The research leading to these results has received funding from the European Research Council under the European Union's Seventh Framework Programme (FP/2007-2013)/ERC Grant Agreement no. 319456. We are grateful to the families who generously supported this trial. The work was also supported by the NIHR Biomedical Research Centre at Guy's and St Thomas' NHS Trust, London, United Kingdom.

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

## Appendix A. *dHCP* image acquisition protocol for fetal MRI

Data was obtained retrospectively from the *dHCP* fetal cohort. A total of 249 T2-weighted scans were acquired by the by the *dHCP* consortium using a Turbo Spin Echo (TSE) sequence on a 3T Philips scanner, following a protocol described by Price and colleagues (Price et al., 2018). Acquisition of fetal MRI at 3T presents various challenges with regards to increased magnetic field inhomogeneity for both B1 and B0; the *dHCP* protocol therefore made use of RF shimming, as well as localized image-based static field shimming (Gaspar et al., 2019). In order to address spontaneous fetal movements and maternal-induced motion, 2D snapshot multi-slice acquisition was used. Acquisition parameters were: TR= 2265 ms; TE= 250 ms; resolution= 1.1 x 1.1 x 2.2 mm. Data was captured from 6 uniquely oriented stacks centered to the fetal brain using a zoomed, multi-band, single-shot TSE sequence. A multi-band, tip-back preparation pulse was used for increased signal-to-noise (SNR) efficiency. Volumetric fetal brain reconstructions were automatically produced from the six stacks (Price et al., 2019). Brain localization was carried out using a deep distance-regression network (Cordero-Grande et al., 2019). Slices were aligned by applying a global search algorithm in the rigid transform space, followed by a multi-resolution registration step that uses a fractional derivative metric. Finally, images were reconstructed using an outliers-robust hybrid 1,2-norm and linear high order regularization approach (Cordero-Grande et al., 2019).

## Appendix B. Preliminary labels generated by *Draw-EM*

In essence, *Draw-EM* employs a two-step process: (i) registration using a spatial prior term, and (ii) segmentation using an intensity model of the image being processed. The spatial prior probability of the different brain structures is defined using an age-specific atlas. The atlas is registered to the target image, and its labels are transformed and averaged according to the local similarity of the atlas. Following this, a Gaussian Mixture Model is used to approximate the intensity model of the image. *Draw-EM* was originally designed for use on neonatal scans and had conventionally used the label-based encephalic ROI template (ALBERTs) atlas by (Gousias et al., 2012). Since we focused on fetal subjects as opposed to neonates, we employed a variant of *Draw-EM* that incorporated the developing brain atlas (Serag, 2013) rather than ALBERTs.

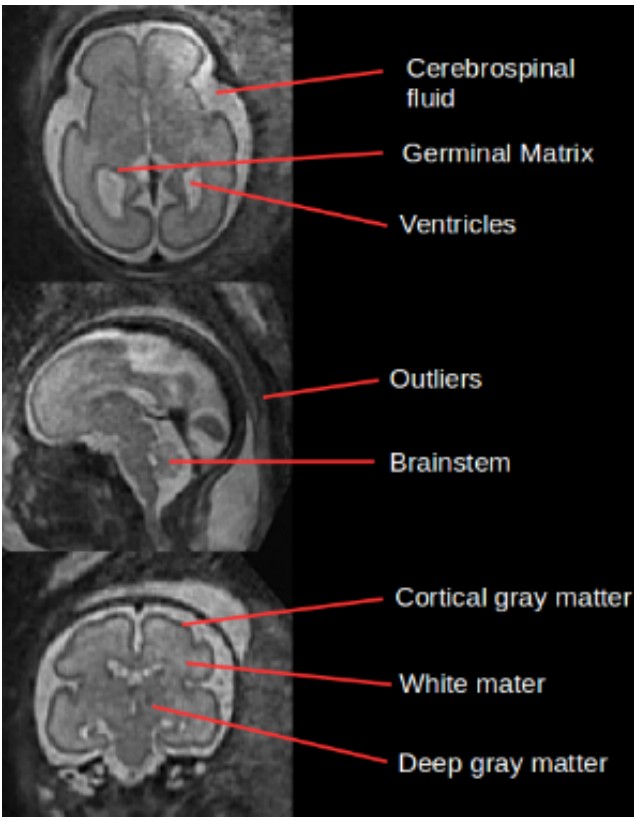

Figure 7: MRI of a fetal subject from the *dHCP* cohort showing various tissue classes. At the first instance, preliminary labels of these tissues were generated using the *Draw-EM* algorithm in order to address the ground-truth bottleneck.

Figures below show T2-weighted scans (GA 31.6 and 24.5 weeks) that passed and failed the quality control step, respectively. Various aspects may have contributed to such discrepancies; we speculate that relatively bright areas (amniotic fluid) surrounding the brain region may have obscured *Draw-EM*'s workflow. This pattern was apparent in other scans

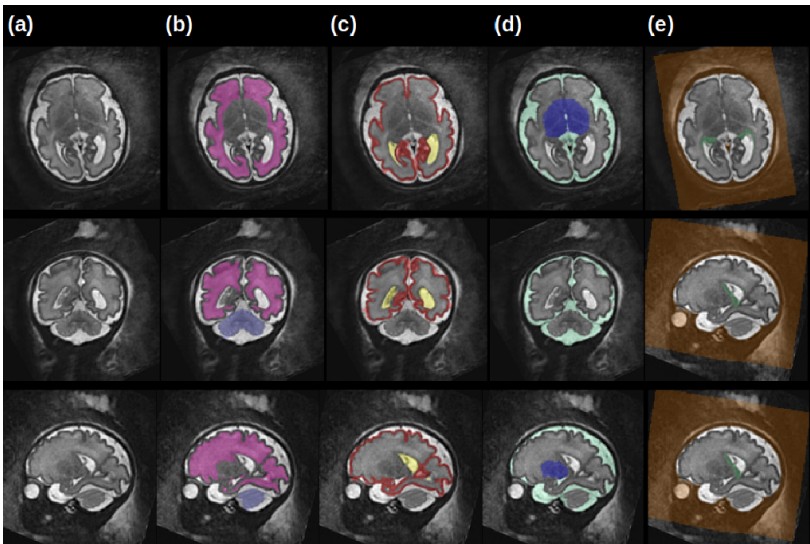

Figure 8: Example T2-weighted scan assigned the *pass* category when processed by *Draw-EM* in the manual quality control step. (a) Mid slices shown in the axial, coronal, and sagittal views. Preliminary tissue labels visualized on top of the slices: (b) white matter and brainstem, (c) cortical gray matter and ventricles, (d) CSF and deep gray matter, and (e) germinal matrix and outlier tissues. The subject was scanned at GA of 31.6 weeks.

that did not result in acceptable tissue labels. The 92 scans processed by *Draw-EM* and assigned the *pass* category were included in subsequent stages for developing the deep segmentation model.

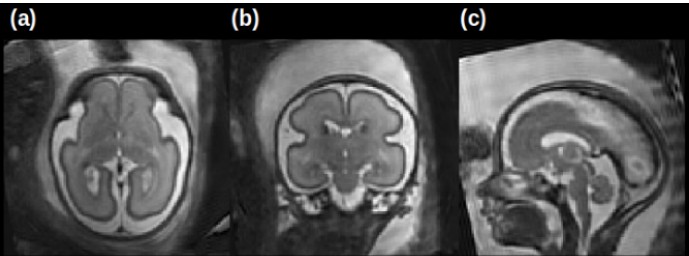

Figure 9: Mid (a) axial, (b) coronal, and (c) sagittal slices of an MRI scan (GA 25.4 weeks) that failed to produce acceptable labels by *Draw-EM*. We speculate that the bright regions surrounding the brain have obscured *Draw-EM*. This pattern was consistent with other scans that failed the manual quality control step.

## Appendix C. Metrics computed on full scans in *held-out-set-C* (n=43) for evaluating the initial segmentation model *SN1*

Table 1: Dice Similarity Coefficient

| Tissue class | Mean | SD |
|---|---|---|
| Brainstem | 0.897 | 0.048 |
| CSF | 0.848 | 0.095 |
| Deep gray matter | 0.877 | 0.048 |
| Germinal matrix | 0.557 | 0.085 |
| *Cortical gray matter* | *0.763* | *0.085* |
| Outlier tissues | 0.900 | 0.037 |
| Ventricles | 0.806 | 0.078 |
| White matter | 0.901 | 0.050 |

Table 2: Sensitivity

| Tissue class | Mean | SD |
|---|---|---|
| Brainstem | 0.899 | 0.055 |
| CSF | 0.866 | 0.077 |
| Deep gray matter | 0.892 | 0.089 |
| Germinal matrix | 0.560 | 0.129 |
| *Cortical gray matter* | *0.744* | *0.099* |
| Outlier tissues | 0.885 | 0.061 |
| Ventricles | 0.842 | 0.099 |
| White matter | 0.919 | 0.052 |

Table 3: Specificity

| Tissue class | Mean | SD |
|---|---|---|
| Brainstem | 1.000 | 0.000 |
| CSF | 0.995 | 0.002 |
| Deep gray matter | 0.999 | 0.000 |
| Germinal matrix | 0.999 | 0.000 |
| *Cortical gray matter* | *0.995* | *0.002* |
| Outlier tissues | 0.943 | 0.036 |
| Ventricles | 1.000 | 0.000 |
| White matter | 0.995 | 0.003 |

Table 4: Positive Predictive Value

| Tissue class | Mean | SD |
|---|---|---|
| Brainstem | 0.902 | 0.085 |
| CSF | 0.836 | 0.117 |
| Deep gray matter | 0.871 | 0.056 |
| Germinal matrix | 0.569 | 0.060 |
| *Cortical gray matter* | *0.787* | *0.078* |
| Outlier tissues | 0.924 | 0.078 |
| Ventricles | 0.790 | 0.116 |
| White matter | 0.890 | 0.072 |

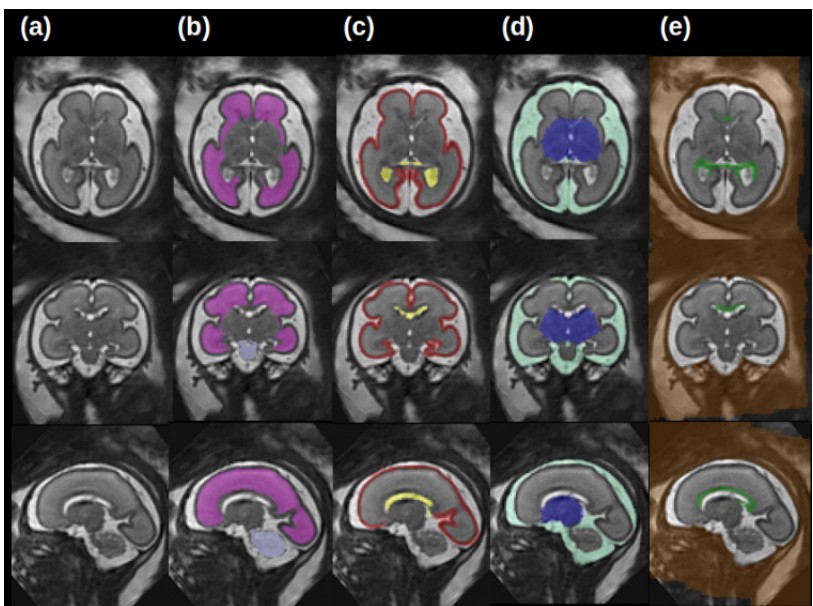

Figure 10: Example performance of *SN1* on an unseen T2-weighted scan of a fetal subject (GA 27.4) in held-out-set-C (a) Mid slices shown in the axial, coronal, and sagittal views. Segmentation performance of initial model *SN1*, overlayed on the slices: (b) white matter and brainstem, (c) cortical gray matter and ventricles, (d) CSF and deep gray matter, and (e) germinal matrix and outlier tissues.

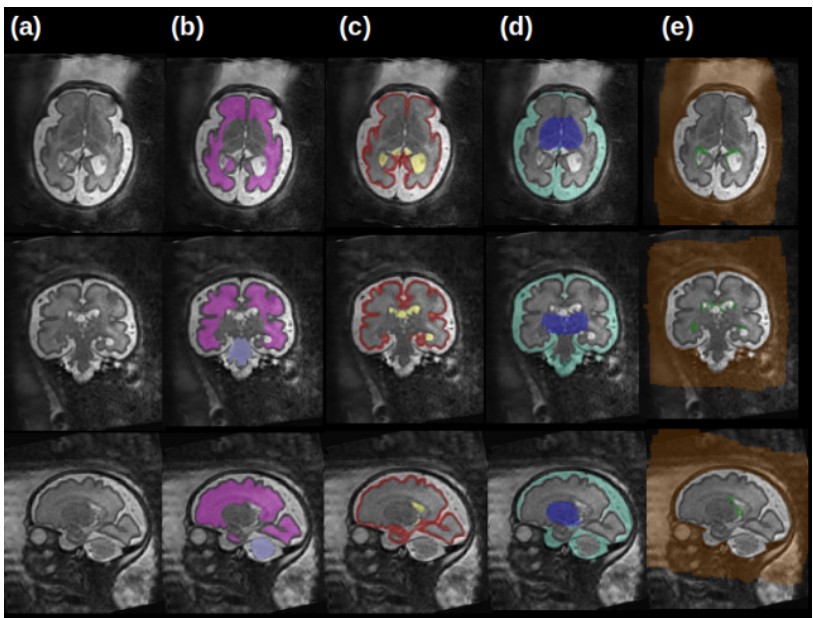

Figure 11: Example performance of *SN1* on an unseen T2-weighted scan of a fetal subject (GA 30.1) in *held-out-set-C* (a) Mid slices shown in the axial, coronal, and sagittal views. Segmentation performance of initial model *SN1*, overlayed on the slices: (b) white matter and brainstem, (c) cortical gray matter and ventricles, (d) CSF and deep gray matter, and (e) germinal matrix and outlier tissues.

## Appendix D. Exploring transferability of *SN2* into a new domain (1.5T)

41 scans were obtained retrospectively from an independent fetal cohort discussed in the work by (Kyriakopoulou et al., 2013). Acquisition was carried out in Hammersmith Hospital (London, United Kingdom) between 2007 and 2011 as part of a study on cortical overgrowth in fetuses with isolated ventriculomegaly. Readers are referred to the work in (Kyriakopoulou et al., 2013) for details on inclusion and exclusion criteria.

Scans were acquired using a Single-Shot Turbo Spin Echo (ssTSE) sequence on a 1.5T Philips scanner. Acquisition parameters were: TR = 15000ms; TE = 160ms; slice thickness = 2.5mm; slice overlap = 1.5 mm. Data was captured in 3 orthogonal planes (4 transverse, 2 coronal, and 2 sagittal acquisitions). Volumetric reconstructions were automatically produced following an approach described in (Jiang et al., 2007). Scans were oriented into standard axial, coronal and sagittal projections, and voxel sizes were interpolated in order to aid display (0.2 x 0.2 x 1.0 mm).

The volumes had associated cortical gray matter annotations previously carried out by a research clinician, as part of the study reported in (Kyriakopoulou et al., 2013). To explore transferability of *SN2* on fetal scans from a different domain distribution, the fine-tuning process described in 4.3 was repeated, albeit with an additional 28 full volumes from the 1.5T cohort (23 training, 5 validation). In terms of testing, 13 additional volumes obtained from the 1.5T cohort, unseen by the system, were also used. Similar to the corresponding training data, the 1.5T validation volumes had cortical gray matter annotations previously carried out by a clinician. Those helped provide a quantitative assessment of the system's performance across the fetal age distribution. A longer training cycle (100 epochs) was used in order to account for the heterogeneity of the data.

Preliminary performance shows mixed success but is encouraging; the model is capable of delivering accurate cortical segmentation, but that performance is not as robust as with the *dHCP*-only system. The figure shows example segmentation on sample slices across various GAs, and the corresponding DSC obtained for the entire volume. It is likely that the lower resolution of the scans obscured the network's performance.

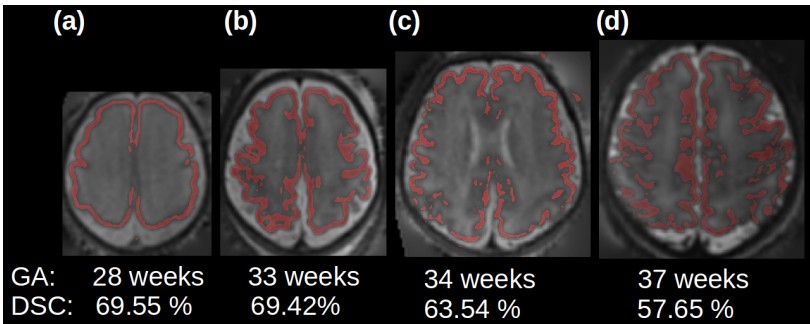

Figure 12: T2-weighted slices and overlayed segmentation masks showing example performance of model on completely held out 1.5 T scans; (a) and (b) presented with Isolated Ventriculomegaly, (c) and (d) did not.

