# OpenReview forum: "A deep learning approach to segmentation of the developing cortex in fetal brain MRI with minimal manual labeling"
_MIDL.io/2020/Conference — MIDL 2020_

### Official Review · AnonReviewer4 · 2020-02-20
**Fetal brain segmentation using a hard to understand data separation**

**Rating:** 2
**Confidence:** 4
**Recommendation:** Poster

**Summary:**

The authors investigate if it is possible to train a CNN for fetal brain segmentation using minimal manual labeling. The authors first train a CNN using labels generated with another program called Draw-EM. The output of this network is manually refined for 283 2D slices. It is unclear to me if this refined data is used to retrain the CNN and how. In addition, the data separation is so complex that the paper is hard to follow. Although the qualitative results shown in Figure 3, 4 and 5 look promising, no qualitative results are given and no comparison with other work is made. This makes it impossible to determine if this approach is valid.




**Strengths:**

The paper has a very nice overview of related work and the authors show promising results. It is clear that a lot of work has gone into this paper. Unfortunately, the data separation makes it almost impossible to understand what the authors did and the experiments do not address the problem.

**Weaknesses:**

There are three main weaknesses:
(1) Very complex data separation. I had to write out what the authors did because it was very hard to understand: 249 T2-weighted scans split into five(!) sets. 249 was split into 151 + 98. 151 were computed using Draw-EM, 59 failed and 92 passed. These 92 passed were split into 39 train(1), 10 validation(2) and 43 named held-out-c (3). The 98 scans were split into 12 scans names held-out-b (4) and 86 scans named held-out-A (5). This is impossible to follow. The authors should just define a training, validation and test set.  Use the training set to train, the validation set to validate, let the expert adjust the result on 283 2D slices in the training set, retrain the network and evaluate on the test set.
(2) The authors train two networks (SN1 and SN2), all the details of these networks are not reported in the paper. In addition, it is unclear to me why SN1 predicts eight classes, while SN2 predicts two classes.
(3) The authors only show qualitative results. The expert should also annotate 2D slices in the test set, so the authors can show that their CNN outperforms the Draw-EM system and the CNN that was only trained using the Draw-EM as input. It remains unclear if this method improves results.


**Detailed Comments:**

paragraph 2.2 mentions that imaging artefacts are caused by fetal and maternal movement, but the authors do not mention how many times these artefacts occur in their dataset. If the used dataset does not contain motion artefacts, please remove this from the paper, because this problem is not adressed. If it includes artefacts, please show that the CNN also works on these cases.
Paragraph 4.2. Please simpify this paragraph.
Paragraph 4.2. ii) the scans were normalized, but did the authors also use a biasfield correction?


**Justification Of Rating:**

Although the authors clearly did a lot of work, the performed experiments are not adequate to answer the research question. MIDL is a deep learning conference, but the paper does not describe which CNN is used and how it was trained.

**Paper Type:**

validation/application paper

**Questions To Address In The Rebuttal:**

Simplify the paper, so it becomes understandable. SN1 is trained on 39 scans and validated on 10. The remaining 200 cases can be the test set. Report how many times Draw-EM fails on these 200 cases (at least 59 times), please report how many times your CNN fails on these 200 cases.
In addition, the expert should also annotate some slices of the test set to get a quantitative result. I would say that it is valid for the expert to adjust the output of Draw-EM. This creates a bias towards Draw-EM, but if your CNN outperforms Draw-EM on the adjusted output, you can show that your CNN is better. If this is not the case, people can better keep using Draw-EM.
Explain why you train two CNN's (SN1 and SN2). If the goal is only to separate cortex vs. non-cortex please simplify the paper and only show these results.
MIDL is a deep learning conference, so please specify your CNN architecture and describe your training methodology.

**Special Issue:**

no

---

> ### Author Response · Authors · 2020-03-26
> **The data separation strategy is needed to ensure no information leakage takes place. We have clarified all the points raised and note that several aspects requested by the reviewer are actually in appendices, we will now move those to the main text**
>
> We thank the reviewer for their critique of our paper and for their observations. The reviewer has raised three interesting areas of discussion on our work, and we believe that our clarifications below fully address those points.
>
> With regards to the first point on data separation, we agree that our strategy is not the simple set-up used on benchmarking data when an algorithm is proposed to the community, where the data is split to a straightforward structure (training, validation, testing) and the work optimises for an evaluation metric. The use of multiple held-out sets was intentionally done to ensure no information leakage would take place at any stage of the workflow (e.g. separating the data used to evaluate preliminary labels from the data used to evaluate the final system); we argue that this should actually be encouraged as good ML practice, particularly in translational research like that described in our study. A key contribution of our work is indeed the more sophisticated approach we present, which we agree might be rather lengthy to follow when one is reading from text. Nevertheless, we reported on full details of our set-up for transparency, and we do not see that re-arranging the data is an effective strategy at this point, as the held-out data is no longer held-out and any new results we report could then be inflated. We can also confirm that the reviewer’s summary of our data separation approach is correct, which suggests that our workflow is understandable. Further, we would like to bring the reviewer’s attention to the detailed flow-diagram in Appendix A which is straightforward to follow. Having now clarified the strengths of our set-up and methodology to the reviewer, we will also ensure the workflow is also clear to the readers by moving the diagram to the main paper text.
>
> In terms of the second point, namely the observations on SN1 and SN2, we would like to bring the reviewer’s attention to Appendix D, which has full technical details on the CNN architectures used in this study as well as how the training process was carried out. We agree that MIDL’s focus is on deep learning and that readers may seek this information while reading the paper, so we will now move it to the main body of the text as well. Additionally, we indeed overlooked to clarify the reason behind the discrepancy in number of classes between SN1 and SN2 (9 vs. 2). This difference is because the processes described on cortical grey matter are parts of a substantially bigger project, and are being replicated with the remaining tissue classes using labels generated by SN1. Our intention was to report on exactly the work that was carried out without omitting key information (e.g. DSC reported in Appendix E would be interpreted differently if we did not mention that it was a multiclass problem), although we agree that at the first instance the use of two networks may appear odd. We will now make sure this is absolutely clear in the paper and update the future work part in order to provide context.
>
> Thirdly is the point on our evaluation of the system. The suggestion about carrying out a final quantitative evaluation is good. We agree, and are currently going through a rigorous Quality Control process to evaluate the system using a clinical protocol. This is, however, a very lengthy process due to the complex nature of fetal neuroimaging data. We are therefore happy to publicly share our model in the meantime (e.g. Github) and to provide videos that show effective segmentation performance to support our claims. The reviewer is right in pointing out that a comparison between our system and Draw-EM seems essential. We would like to reiterate an important point, which is that Draw-EM was designed for use on neonatal (infant) MRI as opposed to the more heterogeneous and complex fetal data analysed in our study. Draw-EM is not a tool that is used in fetal imaging, but one that we used to generate preliminary labels due to similarities between fetal and neonatal brain anatomy. Upon training SN1 using labels generated from Draw-EM, SN1 was actually able to segment data in both the pass category (n=92) and the fail category (n=59). We will now make sure that those points are absolutely clear in the paper.
>
> With regards to the detailed comments on how to improve the text in sections 2.2 and 4.2, we agree and are grateful to the reviewer for pointing these out. We will make sure these changes are made in the final version of the paper in order to improve readability.
>
> Finally, we would like to thank the reviewer again, and we believe that our comments and the changes we will make fully address the points raised. We believe that our work offers a major contribution to the specific area of fetal neuroimaging and we believe that the MIDL community could greatly benefit from our findings, as they are easy to scale to a wide range of medical imaging applications.

---

### Official Review · AnonReviewer1 · 2020-02-25
**Review of Submission 136**

**Rating:** 4
**Confidence:** 4
**Recommendation:** Oral

**Summary:**

Providing a reliable segmentation of the major intracranial compartments in
anatomical MRI of fetus is difficult because the white matter is, dependent
on gestation week and location in the brain, not yet myelinated, thus, inhomogeneous.
This submission describes an approach that uses a convolutional neural network
(CNN) to segment the intracranial compartment into 9 tissue types. The text
focuses on the description of the optimization and evaluation of the method's
performance.


**Strengths:**

The topic of this manuscript is within the scope of the conference and of
potential interest to its audience. Text is straightforward to understand
and without major errors, except where noted below. A considerable
improvement for a difficult problem is described here.


**Weaknesses:**

There are a few, relatively minor issues with the current version that should carefully
be reviewed. This submission is too long, even for a journal submission, but is
missing relevant detail in some sections. Details are given below.

**Justification Of Rating:**

Although no methodological advance is presented here, this manuscript
describes a considerable improvement for a difficult problem. In terms
of depth (and well, length) this submission is rather a journal publication.

**Paper Type:**

both

**Questions To Address In The Rebuttal:**

1. First of all: length. While this 21-page submission formally matches with
the guidelines of this conference, it does so by swapping out relevant information
into six appendices. It is impossible to understand the main text without
referring to the appendices. For a better readability, it would be advantageous
to integrate the appendices into the main text, however, breaking the page limit.
2. The main text is verbose in some sections (e.g., 1, 2.1, 2.2, 4.2), and
might be shortened without loosing relevant information. However, there is
considerable doubt that shortening is possible down to match the limits specified
for the main text. On the other hand, some important information is missing
that are detailed below.
3. Too little reference is given to the work of Moeskops et al., who were
among the first to use a CNN for the segmentation of fetal images. A detailed
comparison appears necessary here.
4. Criteria for selecting 283 slices from 101 volumes are not specified.
Were those manually corrected slices? Slices selected for their high (or low?)
quality?
5. Apparently, the intracranial space is not extracted before segmentation,
and reasons for this decision are little discussed and/or are not convincing.
"Time consuming" can hardly be a reason here. Please, clarify.
6. Appendix D: The relationship between the two-pathway model by Kamnitsas and
the three-pathway model used here are unclear. Consider specifying all detail
necessary to reproduce the model in this text.


**Special Issue:**

yes

---

> ### Author Response · Authors · 2020-03-26
> **We thank the reviewer for highlighting how our work offers a novel contribution to the difficult problem of cortical segmentation in fetal neuroimaging. We are happy to fully address the observations made by the reviewer**
>
> We thank the reviewer for the positive feedback and for appreciating our work, its novelty, and its contribution to the fetal neuroimaging domain. We address the the important points raised below, and we appreciate the reviewer taking the time to evaluate our paper.
>
> 1. Indeed a substantial amount of work went into our study, and hence the overall length of the paper is longer than usual (21 pages), which is why we initially resorted to the use of appendices to ensure that the information remained accessible to the reader. In retrospect, we agree that a number of key information ought to be moved to the main paper text, and we will now do so to improve clarity. Specifically, we will move figures and information in Appendix A and D as per our reply to Reviewer 4.
>
> 2. We will then shorten the main text while keeping key information to make room for the material that will be added from appendices, as well the important points discussed below.
>
> 3. We thank the reviewer for the useful observation with regards to prior work. Indeed, a comparison to the work by Moeskops et al. on fetal imaging seems important and relevant. We will include this information in the main paper text to provide more context for the reader.
>
> 4. We aimed to extract three slices (axial, sagittal, coronal) from each of the 101 volumes. This initially resulted in a total of 303 slices. A visual check of the slices was carried out before being included in subsequent steps. This reduced the number of slices of sufficient quality to 283. We will clarify this in the main text.
>
> 5. Brain region extraction is indeed a standard pre-processing step in fetal neuroimaging structural pipelines. We were simply saying that our results suggest that it is possible to effectively use our tool without needing such a pre-processing step. However, we have developed a separate brain region extraction network that can be used alongside our system to provide accurate ROIs. We will now clarify this in the main text.
>
> 6. The suggestion on clarifying the CNN architectures used and how they relate to the original paper by Kamnitsas et al. is good. In addition to moving details on architectures used to the main paper text, we will now elaborate on this point to ensure clarity for the reader. Further, and on the point of reproducibility, we are happy to publicly share our segmentation model (e.g. via a Github link) and point to it in the main paper text.
>
> We would like to thank the reviewer again for the good feedback, and for appreciating how our work offers a considerable improvement for the difficult problem of fetal brain MRI segmentation. We believe that our answers fully address the good points raised in the feedback.

---

### Official Review · AnonReviewer3 · 2020-03-15
**Simple hybrid strategy to address lack of ground truth data for slice-wise only tissue segmentation from fetal MRI**

**Rating:** 2
**Confidence:** 5

**Summary:**

The paper presents a simple hybrid strategy to deal with the lack of ground truth data. This is applied to a slice-wise tissue segmentation of fetal brain MRI where expert segmentations are missing. The strategy starts by training with the results (in part incorrect, but overall passing visual inspection) from an automatic segmentation method, followed by refining with manual corrections of the segmentations. CNN Segmentation was performed with an existing solution called DeepMedic (2D slice-wise segmentation).


**Strengths:**

- The paper shows that pretraining with automatic segmentations lead to great results that can be improved with limited ground truth data
- The final results (from a slice-wise evaluation viewpoint) seem okay (but are only evaluated on slices from the middle of the volume)

**Weaknesses:**

- 2D: The strategy is based for a 2D segmentation and it would be not straightforward to a 3D strategy as the whole volume would need to be corrected, which is extremely time-consuming.
- 2D segmentations are outdated and should be avoided as they perform badly for the full 3D volume (and only perform well in the middle of the volume).
- Evaluation is done only in the middle of the volume and not throughout.
- This paper basically proposes pretraining with automatic segmentation results, which is a rather straightforward and non-novel type of pretraining, followed by manual segmentations.
- There is little methodological novelty here, and there is relatively little new here

**Detailed Comments:**

No further comments

**Justification Of Rating:**

- Simple strategy for combining pretraining and updating with manual segmentations, no real novelty
- the employed segmentation is uses outdated slice-wise methodology and the corresponding strategy does not easily transfer to volume-wise analysis, as the whole volume would need to be manually corrected

**Paper Type:**

both

**Questions To Address In The Rebuttal:**

No questions need to be addressed

**Special Issue:**

no

---

> ### Author Response · Authors · 2020-03-26
> **The points raised in the review are due to unfortunate misunderstandings of our work. Our approach makes full use of 3D imaging and modelling, and offers an elegant first solution to a complex problem in fetal neuroimaging**
>
> We thank the reviewer for the time taken to review our paper and for the comments. The reviewer has mainly raised three points upon which the review was based. We believe that two of these points are misunderstandings of the technical execution of our work (the assumptions that our model is 2D-based and that our evaluation was carried out slice-wise; both are incorrect). We believe that these misunderstandings have in turn not allowed the reviewer to appreciate the technical value of our seemingly "simple" training strategy, which we see as a major strength and elegance as opposed to a weakness, since our approach has led to very good results in a challenging problem for which no previous solution existed. We explain these points in detail below, and invite the reviewer and AC to cross-reference to our paper.
>
> The first primary concern raised by the reviewer is the major misunderstanding that our method is based on 2D modeling (this was raised four times in the review and as one of the two justifications of rating). This is absolutely not the case. DeepMedic is a fully 3D CNN, both clarified in our text but also easy to identify by consulting the title and context of the cited original work, Kamnitsas et al. "Efficient Multi-scale 3D CNN with fully connected 3D CRF". When carried out on the DHCP fetal data, initial network training was performed on full 3D volumes, as opposed to the individual 2D approach assumed by the reviewer. We then extracted samples of individual 2D slices from segmentations produced by DeepMedic; only those slices were refined by an expert fetal MRI annotator to address the issue of training labels bottleneck (perhaps that is why the reviewer assumed that our model is based on a slice-wise architecture). The manually refined slices were then used to fine-tune a cortical segmentation model, which was also done using a 3D-based architecture. Hence, all segmentations in this paper make full use of the 3D imaging data and modelling. It is very unfortunate that this big misunderstanding may have biased the reviewer to misjudge how technically solid our DL training framework is for this setting.
>
> The second point raised as a major weakness is also a big misunderstanding or an incorrect assumption by the reviewer, that is the "evaluation is done only in the middle of the volume and not throughout". We do not see why the reviewer has this impression. Please see Figures 3, 4 and 5 for visual examples. Our study looked into the segmentation of cortical grey matter, and the slices shown in the main paper text illustrate regions where cortical grey matter tissue is present. We agree with the reviewer that it can be difficult to visualise full volumetric segmentation effectively on paper and would therefore be happy to publicly share our segmentation model (e.g. via a Github link) as well as videos that show effective performance to support our claims. Having clarified this misunderstanding and major concern to the reviewer, we will also easily clarify it to the readers by: (a) adding a single sentence in the evaluation section to stress that all evaluation was done in 3D, and (b) point to published segmentations online.
>
> In terms of novelty, the computational simplicity of our set-up is a major strength rather than a weakness. This is because training labels are a bottleneck which is a key issue that impedes the translation of deep learning into routine clinical practice. Fetal neuroimaging data is highly heterogeneous and is hence tremendously difficult and time consuming to label images at a large scale; the process is very cumbersome even for highly experienced clinical annotators. Therefore, for deep learning to realistically find its way in routine clinical practice, the field ought to move beyond definitions of novelty limited to, say, a new segmentation architecture. In addition to offering a cortical segmentation tool for the fetal neuroimaging community, our work shows that it is feasible to accelerate the deep learning process on highly complex fetal data using effectively very few manually labelled slices, and with minimal manual input.
>
> When justifying the review’s rating, the reviewer argues that our approach “uses outdated slice-wise methodology” and “does not easily transfer to volume-wise analysis”.  As explained above, we did not use a slice-wise approach assumed by the reviewer, rather our model is a fully 3D CNN and works well on full volumetric data with minimal manual input, which is exactly why our work offers a major contribution to the field and we believe that the MIDL community should be aware of our findings.
>
> We are sorry that these misunderstandings may have biased the reviewer to misjudge how technically solid our DL training framework is. We believe that we have now clarified these concerns, and that these clarifications are sufficient for revisiting the review and the work's rating.

---

> > ### Comment · AnonReviewer3 · 2020-03-27
> > **Clarification helps**
> >
> > Thanks for these important clarifications by the authors.
> > Further comments:
> > - Please make sure to clarify in the paper that this is a 3D approach that is used for the segmentation (the reader should not need to go to the bibliography to look at the title of the referenced paper, it should be obviously evident from the writeup)
> > - Please also ensure to clarify in the paper that the real novelty in the strategy is to update the 3D segmentation with corrections on only selected 2D slices (and that the slicing direction does not matter, i.e. that one can use both corrected axial and coronal and sagittal slices?).
> > - while the proposed approach is still rather straightforward and the 3D Deep Medic segmentation is no longer state of the art (due to the rapid nature of CNN technology changes), these clarifications significantly improve my assessment. As the authors mentioned in their response, this shows that their DL framework is solid, but that their writeup needs improvement.
> >
> > This moves my assessment to a "weak accept"

---

> > > ### Author Response · Authors · 2020-03-27
> > > **The authors are thankful and will update the paper**
> > >
> > > We thank the reviewer and we are very glad that these misunderstandings are now resolved.
> > >
> > > We assure the reviewer that we will update the paper with a number of appropriate short sentences to make these points crystal clear. We will also publicly share the model and demos as promised.
> > >
> > > Now that misunderstandings have been resolved, we would greatly appreciate if the new revision of the work's rating is also updated in the corresponding field,  so that other reviewers and readers get the appropriate view of the paper as well.
> > >
> > > We much appreciate the reviewer's time on this.

---

> > > > ### Comment · AnonReviewer3 · 2020-03-27
> > > > **Original review is not editable**
> > > >
> > > > Quick note: all reviewers see all comments, so the other reviewers see my updated assessment. The original assessment itself cannot be edited (at least I cannot see a way how that would be done).

---

### Meta-Review · Area_Chair1 · 2020-04-07
**MetaReview of Paper136 by AreaChair1**

**Rating:** 3

**Metareview:**

Interesting paper on fetal MRI segmentation with minimal labeling requirements. Productive use of the rebuttal period. The reviewers appear largely in agreement after discussion.

**Paper Type:**

validation/application paper

**Special Issue:**

no

---

### Decision · Program_Chairs · 2020-04-11

Accept